# An Efficient Pruning Algorithm for Robust Isotonic Regression

**Cong Han Lim** *
School of Industrial Systems and Engineering
Georgia Tech
Altanta, GA 30332
clim31@gatech.edu

## Abstract

We study a generalization of the classic isotonic regression problem where we allow separable nonconvex objective functions, focusing on the case where the functions are estimators used in robust regression. One can solve this problem to within $\epsilon$-accuracy (of the global minimum) in $O(n/\epsilon)$ using a simple dynamic program, and the complexity of this approach is independent of the underlying functions. We introduce an algorithm that combines techniques from the convex case with branch-and-bound ideas that is able to exploit the shape of the functions. Our algorithm achieves the best known bounds for both the convex case ($O(n \log(1/\epsilon))$) and the general nonconvex case. Experiments show that this algorithm can perform much faster than the dynamic programming approach on robust estimators, especially as the desired accuracy increases.

## 1 Introduction

In this paper we study the following optimization problem with monotonicity constraints:

$$\min_{x \in [0,1]^n} \sum_{i \in [n]} f_i(x_i) \text{ where } x_i \leq x_{i+1} \text{ for } i \in [n-1] \qquad (1)$$

where the functions $f_1, f_2, \ldots, f_n : [0,1] \to \mathbb{R}$ *may be nonconvex* and the notation $[n]$ denotes the set $\{1, 2, \ldots, n\}$. Our goal is to develop an algorithm that achieves an objective $\epsilon$-close to the *global* optimal value for any $\epsilon > 0$ with a complexity that scales along with the properties of $f$. In particular, we present an algorithm that simultaneously achieves the best known bounds when $f_i$ are convex and also for general $f_i$, while scaling much better in practice than the straightforward approach when considering $f$ used in robust estimation such as Huber Loss, Tukey's biweight function, and MCP.

Problem (1) is a generalization of the classic *isotonic regression* problem (Brunk, 1955; Ayer et al., 1955). The goal there to find the best isotonic fit in terms of *Euclidean distance* to a given set of points $y_1, y_2, \ldots, y_n$. This corresponds to setting each $f_i(x)$ to $\|x_i - y_i\|_2^2$. Besides having applications in domains where such a monotonicity assumption is reasonable, isotonic regression also appears as a key step in other statistical and optimization problems such as learning generalized linear and single index models (Kalai and Sastry, 2009), submodular optimization (Bach, 2013), sparse recovery (Bogdan et al., 2013; Zeng and Figueiredo, 2014), and ranking problems (Gunasekar et al., 2016).

There are several reasons to go beyond Euclidean distance and to consider more general $f_i$ functions. For example, using the appropriate Bregman divergence can lead to better regret bounds for certain online learning problems over the convex hull of all rankings (Yasutake et al., 2011; Suehiro et al., 2012), and allowing general $f_i$ functions has applications in computer vision (Hochbaum, 2001;

Kolmogorov et al., 2016). In this paper we will focus on the use of quasiconvex distance functions, the use of which is much more robust to outliers (Bach, 2018)[2]. Figure 1 describes this in more detail.

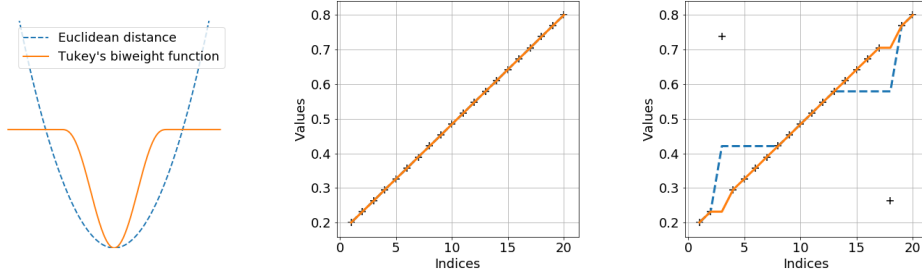

Figure 1: Isotonic regression in the presence of outliers. The left image shows the value of the Euclidean distance and Tukey's biweight function (a canonical function for robust estimation) from $x = -1$ to 1, the middle image demonstrates isotonic regression on a simple linear and noiseless example, and the right image shows how outliers can adversely affect isotonic regression under Euclidean distance.

For general $f_i$ functions we cannot solve Problem (1) exactly (without some strong additional assumptions), and instead we focus on the problem

$$\min_{x \in \mathcal{G}_k^n} \sum_{i \in [n]} f_i(x_i) \text{ where } x_i \leq x_{i+1} \text{ for } i \in [n-1] \tag{2}$$

where instead of allowing the $x_i$ values to lie anywhere in the interval $[0, 1]$, we restrict them to $\mathcal{G}_k := \{0, 1/k, 2/k, \ldots, 1\}$, a equally-spaced grid of $k + 1$ points. This discretized version of the problem will give a feasible solution to the original problem that is close to optimal. The relation between the granularity of the grid and approximation quality for any optimization problem over a bounded domain can be described in terms of the Lipschitz constants of the objective function, and for this particular problem has been described in Bach (2015, 2018) — if functions $f_i$ are Lipschitz continuous with constant $L$, then to obtain a precision of $\epsilon$ in terms of the objective value, it suffices to choose $k \geq 2nL/\epsilon$. One can achieve better bounds using higher-order Lipschitz constants. The main approach for solving Problem (2) for general nonconvex functions is to use dynamic programming (see for example Felzenszwalb and Huttenlocher (2012)) that runs in $O(nk)$. When all the $f_i$ are convex, one can instead use the faster $O(n \log k)$ scaling algorithm by Ahuja and Orlin (2001).

Our main contribution is an algorithm that also achieves $O(nk)$ in the general case and $O(n \log k)$ in the convex case by exploiting the following key fact — *the dynamic programming method always runs in time linear in the sum of possible $x_i$ values over all $x_i$*. Thus, our goal is to limit the range of values by using properties of the $f_i$ functions. This is done by combining ideas from *branch-and-bound* and the scaling algorithm by Ahuja and Orlin (2001) with the dynamic programming approach. When restricted to convex $f_i$ functions, our algorithm is very similar to the scaling algorithm.

Our algorithm works by solving the problem over increasingly finer domains, choosing not to include points that will not get us closer to the global optimum. We use two ways to exclude points, the first of which uses lower bounds over intervals for each $f_i$, and the second requires us to be able to compute a linear underestimate of $f_i$ over an interval efficiently. This information is readily available for a variety of quasiconvex distance functions, and we provide an example of how to compute this for Tukey's biweight function. In practice, this leads to an algorithm that can require far less function evaluations to achieve the same accuracy as dynamic programming, which in turn translates into a faster running time even considering the additional work needed to process each point.

The paper is organized as follows. For the rest of the introduction, we will survey other methods for isotonic regression for specific classes of sets of $f_i$ functions and also mention related problems. Section 2 describes the standard dynamic programming approach. In Section 3, we describe our main pruning algorithm and the key pruning rules for removing $x_i$ values that we need to consider. Section 4 demonstrates the performance of the algorithm on a series of experiments. The longer version of this paper (provided as supplementary material) includes proofs for the linear underestimation rule and also briefly discusses a heuristic variant of our main algorithm.

**Existing methods for isotonic regression.** We will first discuss the main methods for *exactly* solving Problem (1) and the classes of functions the methods can handle. For convex $f_i$ functions, the *pool-adjacent-violators* (PAV) algorithm (Ayer et al., 1955; Brunk, 1955) has been the de facto method for solving the problem. The algorithm was originally developed for the Euclidean distance case but in fact works for any set of convex $f_i$, provided that one can exactly solve intermediate subproblems of the form $\arg\min_{z \in [0,1]} \sum_{i \in S} f_i(z)$ (Best et al., 2000) over subsets $S$ of $[n]$. PAV requires solving up to $n$ such subproblems, and the total cost of solving can be just $O(n)$ for a wide range of functions, including for many Bregman divergences (Lim and Wright, 2016).

There are algorithms for nonconvex functions that are piecewise convex. Let $q$ denote the total number of pieces over all the $f_i$ functions. In the case where the overall functions are convex, piecewise-linear and -quadratic functions can be handled in $O(q \log \log n)$ and $O(q \log n)$ time respectively (Kolmogorov et al., 2016; Hochbaum and Lu, 2017), while in the nonconvex case it is $O(nq)$.

In some cases, we cannot solve the problem exactly and instead deal with the discretized problem (2). For example, this is the case when our knowledge to the functions $f_i$ can only be obtained through function evaluation queries (i.e. $x_i \to f_i(x_i)$). In the convex case, PAV can be used to obtain an algorithm with $O(n^2 \log k)$ time to solve the problem over a grid of $k$ points, but a clever recursive algorithm by Ahuja and Orlin (2001) takes only $O(n \log k)$. A general approach that works for arbitrary functions is dynamic programming, which has a complexity of $O(nk)$.

Bach (2015) recently proposed a framework for optimizing continuous submodular functions that can be applied to solving such functions over monotonicity constraints. This includes separable nonconvex functions as a special case. Although the method is a lot more versatile, when specialized to our setting it results in an algorithm with a complexity of $O(n^2k^2)$. This and dynamic programming are the main known methods for general nonconvex functions.

**Related Problems.** There have been many extensions and variants of the classic isotonic regression problem, and we will briefly describe two of them. One common extension is to use a partial ordering instead of a full ordering. This significantly increases the difficulty of the problem, and this problem can be solved by recursively solving network flow problems. For a detailed survey of this area, which considers different types of partial orderings and $\ell_p$ functions, we refer the reader to Stout (2014). One can also replace the ordering constraints with the pairwise terms $\sum_{i \in [n-1]} g_i(x_{i+1} - x_i)$ where $g_i : \mathbb{R} \to \mathbb{R} \cup \{\infty\}$. By choosing $g_i$ appropriately, we recover many known variants of isotonic regression, including nearly-isotonic regression (Tibshirani et al., 2011), smoothed isotonic regression (Sysoev and Burdakov, 2016; Burdakov and Sysoev, 2017), and a variety of problems from computer vision. The most general recent work (involving piecewise linear functions) is by Hochbaum and Lu (2017). We note that the works by Bach (2015, 2018) also applies in many of these settings.

## 2   Dynamic Programming

We now provide a DP reformulation of Problem (2). Let $A_n^{\mathcal{G}_k}(x_n) := f_n(x_n)$. For any $i \in [n-1]$, we can define the following functions:

$$A_i^{\mathcal{G}_k}(x_i) := f_i(x_i) + C_i^{\mathcal{G}_k}(x_i), \hspace{3cm} \text{(aggregate)}$$

$$C_i^{\mathcal{G}_k}(x_i) := \min_{x_{i+1} \in \mathcal{G}_k} A_{i+1}^{\mathcal{G}_k}(x_{i+1}) \text{ where } x_i \leq x_{i+1}. \hspace{1cm} \text{(min-convolution)}$$

The $A_i^{\mathcal{G}_k}$ functions *aggregate* the accumulated information from the indices $i+1, i+2, \ldots, n$ with the information at the current index $i$, where the $C_i^{\mathcal{G}_k}$ functions represent the *minimum-convolution* of the $A_{i+1}^{\mathcal{G}_k}$ function with the indicator function $g$ where $g(z) = 0$ if $z \leq 0$, and $g = \infty$ otherwise. With this notation, the problem $\min_{x_1 \in \mathcal{G}_k} A_1^{\mathcal{G}_k}(x_1)$ has the same objective and $x_1$ value as Problem (2).

We can use the above recursion to solve the problem, which we formally describe in Algorithm 1. This dynamic programming algorithm can be viewed an application of the Viterbi algorithm. The algorithm does a backward pass, building up all the $A_i^{\mathcal{G}_k}, C_i^{\mathcal{G}_k}$ values from $i = n$ to $i = 1$. Once $A_1^{\mathcal{G}_k}$ has been computed, we know the minimizer $x_1$. We then work our way forwards, each time picking an $x_i$ that minimizes $A_i^{\mathcal{G}_k}$ on the grid $\mathcal{G}_k$ subject to the condition that $x_i \geq x_{i-1}$. The total running time of this algorithm is $O(nk)$, on the order of the number of points in the grid.

---
**Algorithm 1** Dynamic Program for fixed grid $\mathcal{G}_k$
---
    **input:** Functions $\{f_i\}$, Parameter $k$
    $A_n^{\mathcal{G}_k}(z) \leftarrow f_n(z)$ for $z \in \mathcal{G}_k$
    **for** $i = n-1, \ldots, 1$ **do**                                                         ▷ Backwards Pass
       |   $C_i^{\mathcal{G}_k}(1) \leftarrow A_{i+1}^{\mathcal{G}_k}(1)$
       |   $A_i^{\mathcal{G}_k}(1) \leftarrow f_i(1) + C_i^{\mathcal{G}_k}(1)$
       |   **for** $z = {}^{k-1}/k, {}^{k-2}/k, \ldots, 0$ **do**
       |    |   $C_i^{\mathcal{G}_k}(z) \leftarrow \min(A_{i+1}^{\mathcal{G}_k}(z), C_i^{\mathcal{G}_k}(z + {}^1/k))$
       |    |   $A_i^{\mathcal{G}_k}(z) \leftarrow f_i(z) + C_i^{\mathcal{G}_k}(z)$
    $x_0 \leftarrow 0$
    **for** $i = 1, 2 \ldots, n$ **do**                                                             ▷ Forward Pass
       |   $x_i \leftarrow \arg\min_{z \in \mathcal{G}_k, z \geq x_{i-1}} A_i^{\mathcal{G}_k}(z)$
    **return** $(x_1, x_2, \ldots, x_n)$
---

The main drawback of the dynamic programming approach is that it requires us to pick the desired accuracy a priori via choosing an appropriate $k$ value and then overall running time is then $O(nk)$, *no matter the properties of the $f_i$ functions*.

## 3    A Pruning Algorithm for Robust Isotonic Regression

Instead of solving the full discretized problem (2) directly, we can work over a much smaller set of points. Let $x^{\mathcal{G}_k}$ denote an optimal solution to the problem, and for each $i \in [n]$ let $\mathcal{S}_i \subseteq \mathcal{G}_k$ denote a set of points such that $x_i^{\mathcal{G}_k} \in \mathcal{S}_i$. Then

$$\min_{x \in \mathcal{S}_1 \times \ldots \mathcal{S}_n} \sum\nolimits_{i \in [n]} f_i(x_i) \ \text{ where } \ x_i \leq x_{i+1} \text{ for } i \in [n-1],$$

has the same solution $x^{\mathcal{G}_k}$ and it is easy to modify the DP algorithm to work for this problem. All that is needed is to perform the following replacements:

- $z = \ldots$ and $z \in \ldots$ with the appropriate series of points in $\mathcal{S}_i$,

- $C_i^{\mathcal{G}_k}(z_{\max}) \leftarrow \min_{z \geq z_{\max}} A_{i+1}^{\mathcal{G}_k}(z)$ for $z_{\max} = \arg\max(\mathcal{S}_i)$, and

- $C_i^{\mathcal{G}_k}(z) \leftarrow \min(A_{i+1}^{\mathcal{G}_k}(z'))$ where $z' \geq z$.

The values of the $C_i^{\mathcal{G}_k}, A_i^{\mathcal{G}_k}$ functions are the same for both problem formulations on $x^{\mathcal{G}_k}$.

The modified operations can be performed efficiently by maintaining the appropriate minimum values, and this results in an algorithm with a complexity of just $O(|\mathcal{S}_1| + \ldots |\mathcal{S}_n|)$. Our goal is thus to restrict the size of $\mathcal{S}_i$ sets. We perform this by starting from a coarse *set of intervals* $\mathcal{I}_i$ for each index $i$ that initially contains just $[0, 1]$. This contains all points in $\mathcal{G}_k$. We repeatedly subdivide each interval into two and *keep only the intervals that may contain certain better solutions*, which in turn reduces the number of points in $\mathcal{G}_k$ that are contained in some interval.

From here on we assume that $k$ is a power of 2. Algorithm 2 describes the basic framework which we build on throughout this section.

---
**Algorithm 2** Algorithmic Framework for Faster Robust Isotonic Regression
---
    **input:** Functions $\{f_i\}$, Parameter $k$
    $k' \leftarrow 1$
    $\mathcal{I}_i \leftarrow \{[0, 1]\}$ for $i \in [n]$
    **while** $k' < k$ **do**
       |   $\{I_i^{\mathcal{G}_{2k'}}\} \leftarrow$ Refine $\{\mathcal{I}_1^{\mathcal{G}_{k'}}\}$ using $\{f_i\}$
       |   $k' \leftarrow 2k'$
    $x \leftarrow$ run modified DP on endpoints of $\{\mathcal{I}_i^{\mathcal{G}_k}\}$
    **return** $x$
---

At the end of each round of the loop, we want $x^{\mathcal{G}_k}$ be contained in $I_1 \times \ldots \times I_n$ where $I_i$ is some interval from $\mathcal{I}_i$. This ensures that we find the optimal point in the final grid $\mathcal{G}_k$. We also want $\mathcal{I}_i^{\mathcal{G}_k}$ to consist only of intervals of width $1/k'$ with endpoints contained in $\mathcal{G}_{k'}$. This ensures that the overall number of points processed over all iterations is at most $O(nk)$, and by bounding the number of intervals in each $\mathcal{I}_i$ in each iteration we can achieve significantly better performance. In particular, the scaling algorithm for convex functions by Ahuja and Orlin (2001) can be seen as a particular realization of this framework where the refinement process keeps the size of each $\mathcal{I}_i$ to exactly one.

In the rest of this section, we will describe two efficient rules for refining the sets of intervals $\{\mathcal{I}_i\}$ and analyze the complexity of the overall algorithm. The first rule uses lower and upper bounds (akin to standard branch-and-bound), while the second requires one to be able to efficiently construct good linear underestimators of the $f_i$ functions within intervals.

### 3.1 Pruning via lower/upper bounds

This pruning rule constructs lower bounds over the current active intervals, then uses upper bounds (that can be obtained via the aforementioned DP) to decide which intervals can be removed from consideration in subsequent iterations of the algorithm.

We again modify the dynamic program, this time to compute lower bounds over intervals. Let $A_n^{\mathrm{LB},\mathcal{G}_k}(a) := \min_{x_n \in [a, a+1/2^k]} f_n(x_n)$ and recursively define the following:

$$A_i^{\mathrm{LB},\mathcal{G}_k}(a) := \min_{x_i \in [a, a+1/2^k]} f_i(x_i) + C_i^{\mathrm{LB},\mathcal{G}_k}(a), \qquad \text{(aggregate for lower bound)}$$

$$C_i^{\mathrm{LB},\mathcal{G}_k}(a) := \min_{a' \in \mathcal{G}_k} A_{i+1}^{\mathrm{LB},\mathcal{G}_k}(a') \text{ where } a \le a'. \qquad \text{(min-convolution for lower bound)}$$

It is straightforward to see that $A_i^{\mathrm{LB},\mathcal{G}_k}(a)$ is a lower bound for $\sum_{j=i}^{n} f_j(x_j)$ when $x_i$ is contained in the interval $[a, a + 1/2^k]$. This dynamic program can be computed in $O(|\mathcal{I}_1| + \ldots |\mathcal{I}_n|)$ time using the same ideas as before, provided that terms of the form $\min_{x_i \in [a,b]} f_i(x_i)$ can be efficiently calculated.

As for which intervals to keep, we remove an interval $[a, b]$ from $\mathcal{I}_i$ if there is another interval in $\mathcal{I}_i$ which can be used in place of $[a, b]$ and the upper bound from using the other interval is smaller than the lower bound corresponding to $[a, b]$. This concept is formalized in Algorithm 3.

---

**Algorithm 3** Pruning $\mathcal{I}$ via Lower/Upper Bounds

---

**input:** Interval Sets $\{\mathcal{I}_i^{\mathcal{G}_{k'}}\}$, functions $\{f_i\}$, Parameter $k'$
Compute $\{A_i^{\mathcal{G}_{k'}}\}$ and $\{A_i^{\mathrm{LB},\mathcal{G}_{k'}}\}$ using $\{f_i\}$
$Z \leftarrow 0$
**for** $i = 1, \ldots, n$ **do**
$\quad$ $z \leftarrow$ first element in $Z$ sequence
$\quad$ $z' \leftarrow$ next element (1 if there are none)
$\quad$ $\mathcal{J} \leftarrow \emptyset$
$\quad$ **while** $z \ne 1$ **do**
$\quad\quad$ $u \leftarrow \min\{A_i^{\mathcal{G}_{k'}}(x_i) \mid x_i \in \mathcal{G}_{k'} \cap [z, z']\}$
$\quad\quad$ $\mathcal{J} \leftarrow \mathcal{J} \cup \{[a,b] \in \mathcal{I}_i^{\mathcal{G}_{k'}} \mid A_i^{\mathrm{LB},\mathcal{G}_{k'}}(a) \le u, [a,b] \subseteq [z, z']\}$
$\quad\quad$ $z \leftarrow z'$
$\quad\quad$ $z' \leftarrow$ next element in sequence $Z$ (1 if there are none)
$\quad$ $\mathcal{I}_i^{\mathcal{G}_{k'}} \leftarrow \mathcal{J}$
$\quad$ $Z \leftarrow$ all endpoints in $\mathcal{J}$
**return** $\{\mathcal{I}_i^{\mathcal{G}_{k'}}\}$

---

We can show that this procedure does not remove certain solutions, including the optimal solutions to Problems (1) and (2). Definition 3.1 and Proposition 3.2 describes this more precisely.

**Definition 3.1.** *Given a nondecreasing vector $x \in \mathbb{R}^n$, $x$ is $S$-improvable for some $S \subseteq [0, 1]$ if there is a different nondecreasing vector $y \in \mathbb{R}^n$ such that $\sum_{i \in [n]} f_i(y_i) < \sum_{i \in [n]} f_i(x_i)$ and if $y_i \notin S$ it must be the case that $y_i = x_i$.*

Note that the optimal solution $x^{\mathcal{G}_k}$ is not $\mathcal{G}_{k'}$-improvable for any $k'$ that is a factor of $k$.

**Proposition 3.2.** *Let $x^*$ be a nondecreasing vector which is not $\mathcal{G}_{k'}$-improvable. Suppose $x^*$ is in*

$$\prod_{i \in [n]} \left( \bigcup \{[a,b] \in \mathcal{I}_i^{\mathcal{G}_{k'}}\} \right).$$

*This remains true after applying Algorithm 3 to the sets $\{\mathcal{I}_i^{\mathcal{G}_{k'}}\}$.*

## 3.2 Pruning via linear underestimators

We now describe a rule that uses *linear underestimators on intervals in $\mathcal{I}_i$*. In the convex case, one can think of this as using subgradient information. This is what the scaling algorithm of Ahuja and Orlin (2001) uses to obtain a complexity of $O(n \log k)$. We will rely on the following assumption.

**Assumption 3.3.** *Given $a, b, c \in [0,1]$ where $a < b < c$, we can compute in constant time $g_i^L, g_i^R \in \mathbb{R}$ such that $f_i(b) + g_i^L \cdot (a - b) \leq f_i(z)$ for $a \leq z < b$ and $f_i(b) + g_i^R \cdot (c - b) \leq f_i(z)$ for $b < z \leq c$.*

This pruning rule works with any $g_i^L, g_i^R$ that satisfies the condition, but the tighter the underestimator, the better our algorithm will perform. In particular, it is ideal to minimize $g_i^L$ and maximize $g_i^R$. For convex functions, the best possible $g_i^R$ is a subgradient of the function.

Suppose we have the interval $[u, v] \in \mathcal{I}_i^{\mathcal{G}_{k'}}$ for $i \in \{s, s+1, \ldots, t\}$. Our goal is to decide for each $i$ if we should include the intervals $[u, {}^{(u+v)}/2]$ and $[{}^{(u+v)}/2, v]$ in $\mathcal{I}_i^{\mathcal{G}_{2k'}}$. We can do this by taking into account linear underestimators for $f_i$ in each of these two intervals and also by considering which $x_i$ may lie outside of $[u, v]$. Algorithm 4 describes how this can be done.

---

**Algorithm 4** Pruning Subroutine

**input:** $\{f_i\}$, $\{s, s+1, \ldots, t\}$, $a, b, c \in [0,1]$ where $a < b < c$, indices $l, r$
Compute $g_i^L, g_i^R$ (from Assumption 3.3) for $i \in [n]$
$S_t^L \leftarrow g_t^L$
$S_i^L \leftarrow g_i^L + \max(S_{i+1}^L, 0)$ for $i \in \{s, s+1, \ldots, t-1\}$
$I^L \leftarrow \{i \mid i \leq l, S_i^L > 0\} \cup \{i \mid l+1 \leq i < k, k \text{ is first index after } l \text{ where } S_k^L \leq 0\}$
$S_s^R \leftarrow g_s^R$
$S_i^R \leftarrow g_i^R + \min(S_{i-1}^R, 0)$ for $i \in \{s+1, \ldots, t\}$
$I^R \leftarrow \{i \mid i \geq r, S_i^R \leq 0\} \cup \{i \mid k > i \geq r+1, k \text{ is last index before } r \text{ where } S_k^R > 0\}$
**for** $i = s, s+1, \ldots, t$ **do**
$\quad \mathcal{I}_i \leftarrow \emptyset$
$\quad$ if $i \in I^L$, add $[a,b]$ to $\mathcal{I}_i$
$\quad$ if $i \in I^R$ or $\mathcal{I}_i$ is empty, add $[b,c]$ to $\mathcal{I}_i$
**return** $\{\mathcal{I}_i\}$

---

**Theorem 3.4.** *Consider Algorithm 4 and its inputs. Suppose that there is some nondecreasing vector $x^* \in [0,1]^n$ such that $x^*$ is not $\{b\}$-improvable. Let $s, t$ denote the indices where $x_s^*$ and $x_t^*$ are the first and terms of $x^*$ contained in $[a, c]$ respectively. Suppose $l \geq s - 1$ and $r \leq t + 1$. For any $i \in \{s, s+1, \ldots, t\}$, the term $x_i^*$ is contained in one of the intervals in $\mathcal{I}_i$ returned by the algorithm.*

We use Algorithm 4 as part of a larger procedure over the entire collection of interval sets $\mathcal{I}_1, \ldots, \mathcal{I}_n$. This procedure is detailed in Algorithm 5, and refines the set of intervals by splitting each interval into two and running Algorithm 4 on the pair of adjacent intervals.

**Proposition 3.5.** *Suppose the intervals used as inputs to Algorithm 5 are $\{\mathcal{I}_i^{\mathcal{G}_{k'}}\}$ (i.e. all the endpoints are in $\mathcal{G}_{k'}$). Let $x^* \in [0,1]^n$ be a nondecreasing vector that is not $\mathcal{G}_{2k'}$-improvable and is contained in $\prod_{i \in [n]} \left( \bigcup \{[a,b] \in \mathcal{I}_i^{\mathcal{G}_{k'}}\} \right)$. Then, $x^*$ is contained in $\prod_{i \in [n]} \left( \bigcup \{[a,b] \in \mathcal{I}_i^{\mathcal{G}_{2k'}}\} \right)$ where $\{\mathcal{I}^{\mathcal{G}_{2k'}}\}$ are the intervals returned by the algorithm.*

## 3.3 Computing Lower Bounds and Linear Underestimators for Quasiconvex Estimators

For quasiconvex functions, we can compute the lower bound over an interval $[a, b]$ by just evaluating the function on the endpoints $a$ and $b$ (and by knowing what the minimizer and minimum value are).

---

**Algorithm 5** Main Algorithm for Refining via Linear Underestimators

---

**input:** Interval Sets $\{\mathcal{I}_i^{\mathcal{G}_{k'}}\}$, functions $\{f_i\}$
$\mathcal{I}_i' \leftarrow \emptyset$ for $i \in [n]$
**for** $[u,v] \in \bigcup_i \mathcal{I}_i^{\mathcal{G}_{k'}}$ **do**
    **for** each contiguous block of indices $s, s+1, \ldots, t$ in $\{i \mid \mathcal{I}_i$ contains $[u,v]\}$ **do**
        $l \leftarrow \max\{i \mid \exists$ an interval to the left of $[u,v]$ contained in $\mathcal{I}_i^{\mathcal{G}_{k'}}\}$
        $r \leftarrow \min\{i \mid \exists$ an interval to the right of $[u,v]$ contained in $\mathcal{I}_i^{\mathcal{G}_{k'}}\}$
        Update $\{\mathcal{I}_i'\}$ with Alg. 4 with inputs $\{f_i\}, \{s, \ldots, t\}, (a,b,c) = (u, {}^{u+v}/2, v)$, indices $l, r$

**return** $\{\mathcal{I}_i'\}$

---

It is straightforward to compute good linear underestimators for many quasiconvex distance functions used in robust statistics. We will discuss how this can be done for the Tukey biweight function, and similar steps can be taken for other popular functions such as the Huber Loss, SCAD, and MCP.

**Example: Tukey's biweight function and how to efficiently compute good $m$ values.**   Tukey's biweight function is a classic function used in robust regression. The function is zero at the origin and the derivative is $x(1-(x/c)^2)^2$ for $|x| < c$ and 0 otherwise for some fixed $c$.

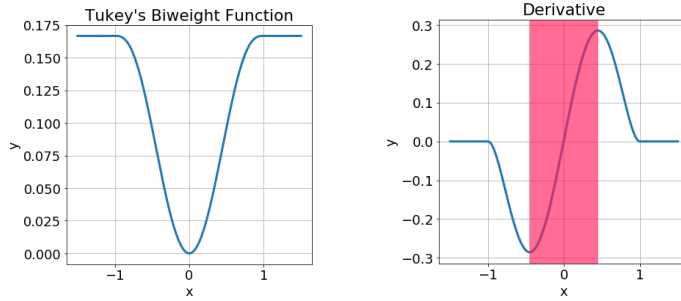

Figure 2: Tukey's biweight function with $c = 1$. In the plot of the derivative, we mark the region in which the function is convex in red ($-1/\sqrt{5} \leq x \leq 1/\sqrt{5}$), while in the other regions at the sides the function is concave.

We will describe how to choose $g^L$ and $g^R$ for $x < 0$, and by symmetry we can use similar methods for $x > 0$. We obtain $g^L$ from connecting $f(x)$ to the largest value of the function. If $x$ is in the convex region, we can simply set $g^R$ to the gradient. We now add a line with slope $-L$ (where $L$ is the largest gradient of the function) to the transition point between the concave and convex regions, and for $x$ in the concave region we obtain $g^R$ by connecting $f(x)$ to this line.

## 3.4   Putting it all together

After stating the pruning and refinement rules for our nonconvex distance functions, we can formally describe in detail the full process in Algorithm 6. The worse case running time is $O(nk)$, since the number of points and intervals processed is on that order and the complexity of the subroutines are linear in those numbers. On the other hand, when the functions $f_i$ are convex,

**Theorem 3.6.** *Algorithm 6 solves Problem* (2) *in $O(nk)$ time in general, and $O(n\log k)$ time for convex functions if we use subgradient information.*

There are two things to note about Algorithm 6. First, it only presents one possible combination of the pruning rules. Another combination would be to not apply the lower/upper bound pruning rule at every iteration. We stick to this particular description in our experiments and theorems for simplicity. Second, we only require the linear underestimator rule for the $O(n\log k)$ convex bound, since that suffices to ensure that sets $\mathcal{S}_i$ have at most a few points.

**Algorithm 6** A Pruning Algorithm for Robust Isotonic Regression

---

**input:** Functions $\{f_i\}$, Parameter $k$

$k' \leftarrow 1$
$\mathcal{S}_i \leftarrow \{0, 1\}$ for $i \in [n]$
$\mathcal{I}_i \leftarrow \{[0, 1]\}$ for $i \in [n]$
**while** $k' < k$ **do**
$\quad \mid \quad \{I_i^{\mathcal{G}_{2k'}}\} \leftarrow$ Algorithm 5 to refine and prune $\{I_i^{\mathcal{G}_{k'}}\}$
$\quad \mid \quad \{I_i^{\mathcal{G}_{2k'}}\} \leftarrow$ Algorithm 3 to prune $\{I_i^{\mathcal{G}_{2k'}}\}$
$\quad \mid \quad k' \leftarrow 2k'$

$x \leftarrow$ run modified DP on endpoints of $\{\mathcal{I}_i^{\mathcal{G}_k}\}$
**return** $x$

---

## 4    Empirical Observations

We evaluate the efficiency of the DP approach and our algorithm on a simple isotonic regression task. We adopt an experiment setup similar to the one used by Bach (2018). We generate a series of $n$ points $y_1, \ldots, y_n$ from 0.2 to 0.8 equally spaced out and added Gaussian random noise with standard deviation of 0.03. We then randomly flipped between 5% to 50% of the points around 0.5, and these points act as the outliers. Our goal now is to test the computational effort required to solve Problem (2). where $f$ is the Tukey's biweight function with $c = 0.3$. We set $n$ to 1000 and varied $k$ from $2^7 = 128$ to $2^{16} = 65536$.

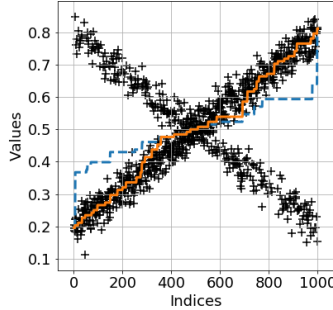

Figure 3:   The $y_i$ points (pluses) and results of using Euclidean distance (blue, dashed) vs. Tukey's biweight function (orange, solid).

We used two metrics to evaluate the computational efficiency. The first measure we use is the total number of points in all $\mathcal{S}_i$ across all iterations, an implementation-independent measure. The second is the wall-clock time taken. The algorithms were implemented in Python 3.6.7, and the experiments were ran on an Intel 7th generation core i5-7200U dual-core processor with 8GB of RAM.

The results are summarized in Figure 4, where the results are averaged over 10 independent trials. In the first figure on the left, we see how the error decreases with an increase in $k$, reflecting the equation that $k \geq O(1/\epsilon)$ is needed to achieve an error of $\epsilon$ in the objective.

In the second and third figures, we compare the performance of the dynamic program against our method, with different percentages of points flipped/corrupted. Instead of presenting three DP lines for each percentage, we simply use one line since the number of points evaluated is always the same and the variation in the timing across all runs is significantly less than 5 percent for all values of $k$. The fact that our method performs differently for different levels of corruption indicates that the performance of our method varies with the difficulty of the problem, a key design goal.

The difference between the second and third figures for our method is approximately a constant factor, indicating that the computational effort for each point is roughly the same. We also see that our method takes significantly more effort per point. Nonetheless our method is significantly faster than the DP across all tested levels of corruption, and the difference gets more significant as we increase $k$.

To more closely investigate how the difficulty of the problem can affect the running time performance, we compare how the speedup is affected by the percent of flipped/corrupted points in Figure 5 at

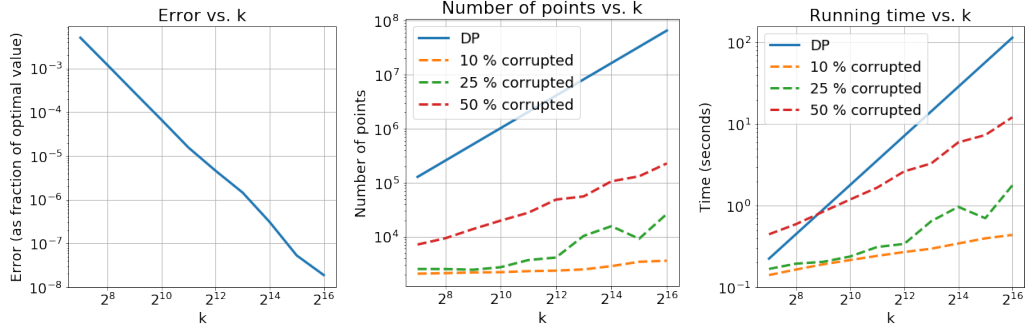

Figure 4: Summary of empirical results. The graph on the left shows how increasing the granularity of the grid decreases the error, and the next two graphs compare the performance of DP against our method (under different percent of flipped/corrupted points) in terms of points processed and running time.

$k = 2^{16}$. For low levels of noise, the speedup is extremely high. There is a rapid decrease in performance between 20 and 30 percent, and at higher levels of noise the performance begins to stabilize again at about 9-10×.

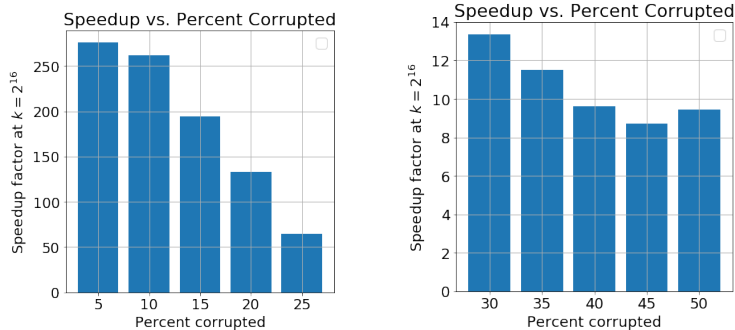

Figure 5: Speedup as a function of the amount of points that were flipped/corrupted at $k = 2^{16}$.

In addition to the above experiments, we also ran preliminary experiments varying the value of $n$. As predicted by the theory, the complexity of both methods scale roughly linearly with $n$. Tests on a range of quasiconvex robust estimators shows similar results.

## 5  Conclusions and Future Work

We propose a pruning algorithm that builds upon the standard DP algorithm for solving the separable nonconvex isotonic regression problem (1) to any arbitrary accuracy (to the global optimal value). On the theoretical front, we demonstrate that the pruning rules developed retain the correct points and intervals required to reach the global optimal value, and in the convex case our algorithm becomes a variant of the $O(n \log k)$ scaling algorithm. In terms of empirical performance, our initial synthetic experiments show that our algorithm scales significantly better as the desired accuracy increases.

Besides developing more pruning rules that can work on a larger range of nonconvex $f_i$ functions, there are two main directions for extensions to this work, mirroring the line of developments for the classic isotonic regression problem. The first is go beyond monotonicity constraints and instead consider chain functions $g_i(x_i - x_{i+1})$ that link together adjacent indices. A particularly interesting case is the one where $g_i(x_i)$ incorporates a $\ell_2$-penalty in addition to the monotonicity constraints in order to promote smoothness. The second is to go from the full ordering we consider here to general partial orders. Dynamic programming approaches fail in that setting and we would require a significantly different approach. It may be possible to adapt the general submodular-based approach developed by Bach (2018), which works in both the above mentioned extensions.

## Acknowledgements

The author would like to thank Alberto Del Pia and Silvia Di Gregorio for initial discussion that lead to this work. The author was partially supported by NSF Award CMMI-1634597, NSF Award IIS-1447449 at UW-Madison. Part of the work was completed while visiting the Simons Institute for the Theory of Computing (partially supported by the DIMACS/Simons Collaboration on Bridging Continuous and Discrete Optimization through NSF Award CCF-1740425).

## Footnotes

*Work done while at Wisconsin Institute for Discovery, University of Wisconsin-Madison.

[2] Our focus in this paper is on developing algorithms for global optimization. For more on robust estimators, we refer the reader to textbooks by Huber (2004); Hampel et al. (2011).

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
