[Supplementary Material]

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

# A Proofs for the Linear Underestimator Pruning Rule

## A.1 Elementary facts on sequences on numbers

The following simple results on sequences of numbers will be useful in analyzing the correctness of the algorithm. These facts are elementary and we include the proofs for completeness.

**Lemma A.1.** *Let* $x_1, x_2, \ldots, x_n$ *be a sequence of real numbers, and let* $k$ *be the largest number that maximizes* $\sum_{i \in [k]} x_i$, *where* $k$ *is allowed to be zero.*

1. *For any subsequence* $T = t + 1, t + 2, \ldots, k$ *for some* $t \geq 0$, $\sum_{i \in T} x_i$ *is greater than zero.*

2. *If we extend the sequence, the new maximizer* $k''$ *is greater than or equal to* $k$.

3. *Let* $y_1, y_2, \ldots, y_n$ *denote a sequence of positive real numbers. Then the largest number* $k'$ *that maximizes* $\sum_{i \in [k']} (x_i + y_i)$ *satisfies* $k' \geq k$.

4. *Let* $\lambda_1, \lambda_2, \ldots, \lambda_n$ *be an increasing positive sequence. Then the largest number* $k''$ *that maximizes* $\sum_{i \in [k'']} \lambda_i x_i$ *satisfies* $k'' \geq k$.

*Proof.* The first claim is true since if there is some $t$ where the corresponding sum is less than zero, then $\sum_{i \in [t]} x_i > \sum_{i \in [k]} x_i$, contradicting the maximality of using $k$.

The second and third claims hold since any shorter sequence is dominated by the original sequence.

To prove the fourth claim, we suppose for contradiction that some $s < k$ maximizes the sum $\sum_{i \in [s]} \lambda_i x_i$. Our first claim implies $x_s$ has to be greater or equal to zero, or that $s = 0$. It also implies $\sum_{i \in \{s+1, s+2, \ldots, k\}} x_i \geq 0$.

If $\sum_{i \in \{s+1, s+2, \ldots, k'\}} \lambda_i x_i$ where $k' \leq k$ is positive, this will contradict the maximality of using the index $s$. It now suffices to prove the following claim:

> Suppose $\sum_{i \in [n]} x_i \geq 0$ and $\sum_{i \in [m]} x_i < 0$ for all $m \in [n-1]$. Let $\lambda_1, \lambda_2, \ldots, \lambda_n$ be an increasing positive sequence. Then there exists $k' \in [n]$ such that $\sum_{i \in [k']} \lambda_i x_i \geq 0$.

We will prove this by induction. The $n = 1$ case is trivial. Now suppose the claim is true up to some $n'$. Let $m'$ denote the first index where $x_{m'} \geq 0$. If $m' = 1$, we are done since we can set $k'$ to $m'$.

Now suppose $m' > 1$. Consider the sequence $y_1, y_2, \ldots, y_{n'-m'+2}$, where

$$y_1 = \frac{1}{\lambda_{m'}} \sum_{i \in [m']} \lambda_i x_i,$$

$y_2 = x_{m'+1}, \ldots, y_{n'-m'+2} = x_{n'+1}$. Note that $y_1 \geq \sum_{i \in [m']} x_i$ from the fact that $x_{m'} \geq 0$ while all earlier terms are negative and have been scaled by a smaller amount. This means that $\sum_{i \in [n'-m'+2]} y_i$ is greater than equal to zero. Hence, we can pick some smallest index $n'' \leq n' - m' + 2 \leq n'$ where $\sum_{i \in [n'']} y_i \geq 0$.

Consider the increasing sequence of coefficients $\mu$ where $\mu_i = \lambda_{m'+i-1}$. By the inductive hypothesis, we can find some index $p$ where $\sum_{i \in [p]} \lambda_i y_i \geq 0$. Note that

$$\sum_{i \in [p]} \lambda_i y_i = \sum_{i \in [p-m'+1]} \lambda_i x_i$$

which concludes the proof. $\qquad\square$

We will now describe how to compute the maximizing index $k$ in Algorithm 7. Note that $D_i$ indicates the best possible sum you can get when starting from index $i$ (insead of index 1). The correctness proof follows from elementary dynamic programming arguments.

**Lemma A.2.** *Algorithm 7 returns the index* $k$ *such that* $\sum_{i \in [k]} x_i$ *is maximized.*

**Algorithm 7** Computing the most positive sequence.

---
**input:** $x_1, x_2, \ldots, x_n$
$D_n \leftarrow x_n$
$D_i \leftarrow x_i + \max(D_i + 1, 0)$ for $i \in [n-1]$
**return** $i - 1$ where $i$ is the index of the smallest negative $A_i$

---

Building on the previous algorithm, we have the following algorithm (Algorithm 8) that will be used as a subroutine in our main linear underestimator-based pruning algorithm. The proof for Proposition A.3 is very similar to the earlier lemma.

---
**Algorithm 8** All most positive sequences starting before or at $m + 1$.

---
**input:** $x_1, x_2, \ldots, x_n$, index $m$
$D_n \leftarrow x_n$
$D_i \leftarrow x_i + \max(D_{i+1}, 0)$ for $i \in [n-1]$
$I \leftarrow \{i \mid i \leq m, D_i \geq 0\}$
**return** $I \cup \{i \mid m + 1 \leq i < k, k \text{ is the first index after } m \text{ where } A_k < 0\}$

---

**Proposition A.3.** *Let $x_1, \ldots, x_n$ and index $m$ be the input to Algorithm 8. Consider an index $m' \leq m$, and let $k'$ be the index that maximizes the sum $\sum_{i \in \{m'+1, \ldots, k'\}} x_i$. Then all indices $m' + 1, \ldots, k'$ are returned by the algorithm.*

As a direct consequence of Lemma A.1 and Proposition A.3, we have that adding positive terms to $x$ or scaling $x$ by a positive monotonically-increasing vector (à la Lemma A.1 third and fourth claims) only increases the set of indices returned.

## A.2  Main proof

We first study a variant of Algorithm 5 that only uses linear underestimation information to the left (i.e. $g^L$). Consider $0 < a < b < 1$.

---
**Algorithm 9** Linear Underestimate Pruning (Left)

---
**input:** $\{f_i\}$, $a, b \in (0, 1)$ where $a < b$, index $m$
Compute $g_i^L$ (defined in Assumption 3.3) for $i \in [n]$
**return** Algorithm 8 on $g_1^L, \ldots, g_n^L$ and index $m$

---

**Definition A.4.** *Given a nondecreasing vector $x \in \mathbb{R}^n$, we say $x$ is S-improvable for some set $S \subseteq [0, 1]$ if there is a nondecreasing vector $y \in \mathbb{R}^n$ such that if $y_i \notin S \Rightarrow y_i = x_i$ and $\sum_{i \in [n]} f_i(y_i) < \sum_{i \in [n]} f_i(x_i)$.*

**Proposition A.5.** *Suppose that there is some nondecreasing vector $x^* \in [0, 1]^n$ such that $x^*$ is not $\{b\}$-improvable. Let $s, t$ denote the indices where $x_s^*$ and $x_t^*$ are the first and terms of $x^*$ contained in $[a, b]$ respectively. Let $T = s, s + 1 \ldots, t$.*

*Then, the output of Algorithm 9 initialized with index $m \geq s - 1$ includes all indices $s, s + 1 \ldots, t$.*

*Proof.* Let $y_i := f_i(b) - f_i(x_i^*)$.

We focus our attention on the indices $T$ for now. We claim that having $k = t$ maximizes the sum $\sum_{i \in \{s, s+1, \ldots, k\}} y_i$ (when $k$ is allowed to be anything from $s - 1$ to $t$). This is because if $k < t$, then it must be the case that $\sum_{i \in \{k+1, \ldots, t\}} y_i < 0$, and this violates the fact that the vector $x^*$ is assumed to be not $\{b\}$-improvable.

Let $\lambda_i := (b - x_i^*)^{-1}$ for $i \in T$. This is an nondecreasing sequence on $T$, so by claim 4 of Lemma A.1, $\sum_{i \in \{s, s+1, \ldots, k\}} \lambda_i y_i$ is maximized again by setting $k = t$. Furthermore, since $g_i^L$ are linear underestimators, we must have

$$g_i^L \geq \frac{f_i(b) - f_i(x_i^*)}{b - x_i^*}$$

which by claim 3 of Lemma A.1 means that again the sum $\sum_{i \in \{s,s+1,\ldots,k\}} g_i^L$ is maximized by setting $k = t$.

We now consider the entire range $[n]$. By claim 2 of Lemma A.1, the index $k$ that maximizes the sum $\sum_{i \in \{s,s+1,\ldots,k\}} y_i$ must satisfy $k \geq t$. Using Proposition A.3, we know that Algorithm 8 will return all indices $s, s+1, \ldots, t, \ldots, k$. □

We note that the main subroutine for linear underestimators (Algorithm 4) is Algorithm 9 combined with a variant of itself that works on the right. The variant now uses $g_i^R$ terms, and performs Algorithm 8 on the sequence $h_1, h_2, \ldots, h_n$ where $h_i = -g_{n-i+1}^R$. This exploits the fact that if $x_1, \ldots, x_n$ is nondecreasing, then so is $-x_n, -x_{n-1}, \ldots, -x_1$. Hence, this implies Theorem 3.4.

# B  A Fast Heuristic Pruning Rule

Instead of using Algorithm 8 as a subroutine in Algorithm 9, we can instead use Algorithm 7. The result is no longer guaranteed to be optimal, but we often recover a solution that is close or equal to the true solution. In all our experiments, the total $\ell_1$ difference is always less than 1. The amount of difference is extremely dependent on the instance, and in the majority of instances at lower noise levels there is no difference.

Figure 6: Summary of empirical results for the heuristic, averaged over 10 instances. The graphs compare the number of points processed and running time against the other methods. Note that the amount of work needed per point is less than the amount needed for our exact method.