[Reviews · NeurIPS 2018]

Reviewer 1



This paper presents an algorithm for efficient pruning in robust isotonic regression. The authors present the method and some theoretical results about the complexity of the algorithms . The algorithm is based on combining ideas from branch-and-bound and the scaling algorithm with the dynamic programming approach. The paper is well written and methods are sufficiently detailed. Although more examples and justifications of the decision could help an inexpert reader to follow the paper. The authors expose that their algorithm is able to obtain good results with lower complexity with respect to related methods. Are there any disadvantage associated to this improvement in resources? A discussion about this could be useful. Section 4 (empirical observations) should be improved. It could be interesting to see the performance of similar algorithms as well as other settings in the experiments .

Reviewer 2



This paper studies fast algorithms for isotonic regression under general loss functions. The author(s) build upon an existing dynamic programming (DP) approach by suggesting two main strategies to prune points -- one based on pruning with lower/upper bounds based on the Lipschitz constants of the loss functions, and another based on "linear underestimators" (which are assumed to be computable in constant time). Overall this is a well written paper. There are minor language issues and ommisions which can be ironed. A few examples are line 69 what is S in argmin_{z in [0,1]} \sum_{i \in S} f_i(z) line 181 the tighter "the" underestimator .. line 240 "and run" should be "and ran" In my opinion the proofs presented in the paper appear a bit heuristic and "hand-wavy", so one suggestion that I have is to strengthen the proofs. One thing that does not become very clear from the paper is the following: does the DP have complexity O(nk) in the general case, and O(n log k) in the convex case (or only a special variant of it has O(n log k) in the convex case)? If so, then the improvement of this paper is not in improving those orders, but is in improving the constants of the orders? Another related question is does the DP require the loss functions to be Lipschitz, as required by the pruning algorithm? If not that would be a disadvantage to the pruning algorithm which seems to rely on the Lipschitz assumptions (and is hence not applicable to the Euclidean distance which is the classic isotonic regression). Finally, I think that including back the omitted plot "running time vs k" would be beneficial since one would see how much real time it takes to run the algorithm.

Reviewer 3



This paper develops algorithms for isotonic regression with non-convex objectives, generalising the usual Euclidean distance. The main idea is to approximate the optimal solution through the use of a set of grid points, and then successively pruning those. I think the work is solid and sound but perhaps of interest to a fairly limited audience only. A few typos: Last sentence of Section 1: "the works by Bach Bach ... applies" Caption of Fig.2: the second "convex" should be "concave".

Reviewer 4



Report on "An Efficient Pruning Algorithm for Robust Isotonic Regression" The paper deals with the problem of isotonic regression and proposes efficient strategies for solving the non convex optimisation problem efficiently, by applying dynamic programming on a sparse grid of x values, appropriately chosen. In practice the proposal boils down in being more efficient in computational time with respect to other methods. Theoretical results and examples are provided to support the idea of the paper. However, there is no comparison on real-life data. Overall the paper is well-written and the results seem sound. My, perhaps obvious questions/comments are: (a) I read this paper as "we propose a method for solving a problem that has practical applications", but the authors have shown us some theoretical results and a simulation study but no application on real-life data is provided. I believe that an application to data is needed in order to fully convince the reader that the method is practically useful. (b) Could you say something about higher dimensions? Is it applicable to two-dimensional problems? Does it scale with the dimensionality of y? Minor issues - in the Abstract, I found the sentence "A simple dynamic programming approach allows us to 4 solve this problem to within ?-accuracy (of the global minimum) in time linear 5 in 1/? and the dimension" difficult to read and understand at first read. - l. 35 "a equally-spaced" should read "an equally-spaced". - l. 53 "a interval efficiently" should read "an interval efficiently."